# The Longitudinal Association Between Habitual Smartphone Use and Peer Attachment: A Random Intercept Latent Transition Analysis

**DOI:** 10.3390/ijerph22040489

**Published:** 2025-03-25

**Authors:** Haoyu Zhao, Michelle Dusko Biferie, Bowen Xiao, Jennifer Shapka

**Affiliations:** Department of Educational and Counselling Psychology, and Special Education, The University of British Columbia, Vancouver, BC V6T 1Z4, Canada

**Keywords:** adolescent, peer attachment, habitual smartphone use

## Abstract

Although many peers socialize online, there is evidence that adolescents who spend too much time online are lonely, depressed, and anxious. This study incorporates habitual smartphone use as a new way of measuring smartphone engagement, based on the shortcomings of simply measuring ‘hours spent online’. Drawing on a large 2-year longitudinal study, the current research aims to investigate the association between habitual smartphone use and peer attachment among Canadian adolescents. A whole-school approach combined with a convenience sampling method was used to select our sample. A total of 1303 Canadian high school students (Grades 8–12; mage = 14.51 years, SD = 1.17 years; 50.3% females) who completed both waves of data collection were included in this study. A random intercept latent transition analysis (RI-LTA) was utilized to assess the association between habitual smartphone use (absent-minded subscale of the Smartphone Usage Questionnaire) and transition probabilities among profiles of peer attachment (Inventory of Parent and Peer Attachment), after adjusting for age, gender, ethnicity, stress, family attachment, school connectedness, and social goals. Three profiles of peer attachment were identified: (Profile 1: weak communication and some alienation; Profile 2: strong communication, strong trust, and weak alienation; Profile 3: okay communication and high alienation). Results of multivariable RI-LTA indicated that increased habitual smartphone use was significantly associated with a heightened probability of transitioning from Profile 2 at Wave 1 to Profile 1 at Wave 2 (odds ratio (OR) = 1.21, 95% confidence interval (CI) 1.003–1.46). Findings indicate that adolescents who are more habituated to their phones may become less attached to their peers over time. This offers insights for caregivers to focus on management and discussing smartphone engagement with adolescents to strengthen their attachment with peers.

## 1. Introduction

Smartphones have become increasingly indispensable and prevalent in our current society. Data have suggested that 95% of adolescents from the United States between 13 and 17 use smartphones, and 46% are online almost all the time [1]. Data from Canadian adolescents has shown similar patterns, with over 30% of them reporting that they use social media consistently throughout the day [2]. Taken together, these findings suggest that even during the day adolescents are interacting with their peers online, suggesting that peer relationships are established and flourish as much online as in offline interactions [3]. While the need for research on how peer relationships are impacted by smartphone use is acknowledged, most current investigations focus on smartphone addiction among adolescents. Although the negative association between smartphone addiction and peer relationships has been well-established among adolescents [4,5,6], habitual smartphone use, a less studied behavior, might be more reflective of smartphone engagement among teenagers. Habitual smartphone use has different features than pathological addictive smartphone use that may implicate factors beyond the typical pathways of addiction (uncontrollable use and distress upon withdrawal). Oulasvirta et al. (2012) argue that habitual smartphone use is unique from smartphone addiction because it is under the person’s control and could lead to positive outcomes such as multitasking skills [7]. Smartphone addiction, however, appears to be outside of one’s control and has been linked to several maladaptive outcomes [8,9,10]. It can be defined as a type of behavioral addiction that involves compulsive and excessive use of smartphones despite the substantial harmful emotions individuals experience, such as guilt, shame, and anxiety [11]. Fortunately, the prevalence of smartphone addiction is relatively low (23%) [12]. However, this low prevalence suggests that we should shift our attention from smartphone addiction to habitual smartphone use to better understand the amount of time youth are spending online. As such, the current study focuses on habitual smartphone use and its impact on subsequent peer attachment, while controlling for other potential risk factors, including stress levels, family attachment, school connectedness, and social goals.

### 1.1. Peer Attachment and Its Predictors

Peer attachment takes on increased significance as children transition to adolescence, a time when adolescents re-navigate their relationships with parents and authority [13]. Strong attachment to psychologically healthy peers has been shown to be associated with various benefits, such as higher academic performance [14] and better mental health [15]. Given the fact that many factors can impact peer attachment, for the current study, we included the following covariates so that we can more precisely explore the impact of habitual smartphone use.

#### 1.1.1. Stress

Stress refers to the inability to cope with external demands called stressors and is understood as an immediate response to stressors. This definition differs from anxiety, which is more future-oriented and not linked to any one specific cause [16]. Previous work has demonstrated that for female adolescents, stress from negative life events is associated with poor peer relationships [17]. Additionally, peer relationship issues such as bullying and less support are associated with elevated stress levels [18]. Given this, the current study also includes stress as a covariate.

#### 1.1.2. School Connectedness

School connectedness gives students a sense of belonging at school and includes how supported they feel by their peers and teachers. High school connectedness implies that students feel they are important to their school and that they are cared for by the school community [19]. Current evidence has suggested that peer relationships are positively associated with school connectedness [20]. Better school connectedness has also been found to be associated with less peer bullying, which could subsequently lead to stronger peer attachment [21]. Therefore, the impact of school connectedness on peer attachment is considered in this study.

#### 1.1.3. Family Attachment

Although attachment in adolescence expands from parents to other significant figures, such as peers, family attachment, being close or connected to one’s family, has still been shown to play an essential role in adolescent socioemotional development [22,23]. More specifically, strong family attachment has been found to generate benefits for adolescents, among which increased peer attachment is a substantial one [24,25]. Thus, family attachment was also included in our analysis.

#### 1.1.4. Social Goals

During adolescence, obtaining a high peer status is an important goal. Social popularity goals refer to the desire to be popular and dominant among peers, while social preference goals refer to being liked and accepted by peers [26]. These two goals are distinct dimensions and can lead to different outcomes, where striving to be popular is linked to more bullying and poorer peer attachment [27,28], whereas prioritizing social reference goals is associated with prosocial behaviors and close peer relationships [29]. Therefore, both social goals were considered in our analysis to indicate adolescents’ online socialization tendencies.

### 1.2. Habitual Smartphone

Cognitive psychology theory defines habits as “an automatic behavior triggered by situational cues” [30,31]. Drawing on the uses and gratifications perspective, smartphone uses can be categorized into (1) instrumental uses, where people use phones for specific purposes, and (2) habitual uses, linked with passing time [32]. Based on the aforementioned definitions, habitual smartphone use means automatically using smartphones without conscious thinking and self-instruction to pursue gratifications and can be triggered by external (e.g., situations) or internal cues (e.g., emotional states) [11,33]. This automaticity also refers to the occurrence of smartphone use that does not require clear goals for using or much cognitive cost. In addition to automaticity, frequency is another aspect of habitual smartphone use. A higher level of automaticity indicates a higher habit strength, which can make use more easily triggered and more frequent [34]. Studies have shown that ‘checking behavior’ also takes up a significant part of habitual smartphone use. More generally, smartphone checking behaviors typically encompass fast processing, as well as short and rapid sessions on the smartphone, often repeated many times during the day [7]. Habitual smartphone use provides a gateway for users to have quick access to rewards (e.g., news notifications or likes from peers), to overcome boredom, and to discover interesting content/information [7,34]. One cross-sectional study has shown that being female and younger and both process and social use of smartphones were associated with habitual smartphone use [11].

### 1.3. Habitual Smartphone Use and Peer Attachment

Rooted in the displacement and stimulation hypothesis [35,36], current research has found contradicting evidence concerning the relationship between smartphone use and peer attachment. For example, the displacement hypothesis suggests that spending time on smartphones can reduce offline interactions with people, which subsequently undermines peer attachment. In contrast, the stimulation hypothesis indicates that peer attachment is positively associated with smartphone use. People get more chances to share with other people online and thus can strengthen their peer attachment by using smartphones. Unfortunately, there is a dearth of research on habitual smartphone use and peer attachment. As noted above, most of the research in this area has focused on problematic smartphone use. Prior cross-sectional work has shown that smartphone addiction is associated with poor peer attachment among different ethnic samples, including Turkish [4], Spanish and Mexican [5], and Chinese adolescents [6]. Similar longitudinal associations were also found for Korean [37] and Dutch adolescents [38].

Screen time as an indicator of smartphone use has also been studied [39,40], but research targeting adolescents is limited. One cross-sectional study suggested that increased screen time strengthened peer attachment for a sample of American early adolescents [41]. Other studies have explored how the purpose of smartphone use impacts peer relationships. One cross-sectional study with Chinese adolescents found that interpersonal interaction use of smartphones had no impact on peer attachment; however, recreational use and online consumption were negatively associated with peer attachment. In contrast, using smartphones for studying or checking health information led to stronger peer attachment [42]. Likewise, increased interactive social media use was found to lead to better peer attachment [43,44,45]. However, none of the smartphone addiction, screen time, or purposes of smartphone use reflect prevalent habitual smartphone use well. Moreover, there were methodological issues with these studies, which exclusively used mean index, sum scores of items, or an absolute number (e.g., number of close friends) to indicate peer attachment in the analysis; thereby failing to capture the complexity of this concept [4,41,44].

### 1.4. Current Study

Given the large amount of time adolescents are engaging in habitual smartphone use, the current study seeks to address an important gap in the literature by examining its impact on adolescent friendships. Specifically, this study is guided by the following overarching research question: How does habitual smartphone use influence adolescents’ friendships over time? To investigate this, we conceptualize habitual smartphone use as the frequency of absent-minded smartphone behaviors—instances where adolescents check their phones automatically, without conscious intent or specific purposes. This study leverages two-wave population-based longitudinal data, enabling us to explore the relationship between habitual smartphone use and peer attachment transitions over time.

To systematically examine this relationship, we define the following research objectives:

(1) To assess whether habitual smartphone use predicts latent peer attachment profile transitions across waves. (2) To examine this relationship after accounting for demographic factors and (3) to investigate the prediction of peer attachment transitions after controlling for all potential covariates. Grounded in existent literature, we hypothesize that adolescents who report higher levels of habitual smartphone use would lead to a transition from stronger peer attachment to poorer peer attachment profiles, either in univariable analysis, after controlling for demographics, or all potential covariates.

## 2. Materials and Methods

### 2.1. Procedures

Data were collected during the school year 2021 (Wave 1) and 2022 (Wave 2). Participants were recruited from five public schools across two public school districts in British Columbia using a whole-school approach and a combination of convenience and snowball sampling. To ensure a representative sample from schools, passive parental consent was utilized. All participants were required to provide active informed assent before participating in the study. That said, adolescents who provided active informed assent and whose parents did not refuse their kids to participate were included in the study. Adolescents who were in grade 12 or categorized as “other sex” in Wave 1 were excluded from our analysis. As a thank you for participating, all potential participants were included in a draw to win a gift certificate. Participants completed questionnaires during school time on their own devices through Qualtrics (www.qualtrics.com (accessed on 23 March 2025)). The study was approved by the institutional ethics review board of the University of British Columbia.

### 2.2. Measures

#### 2.2.1. Demographics

Participants’ self-reported demographic information, including age, sex (female and male), and ethnicity (Asian, White, and other), was used in all analyses.

#### 2.2.2. Peer Attachment

Peer attachment was measured using the previously validated Inventory of Parent and Peer Attachment (IPPA-SV) [46]. The scale consists of 12 items rated on a 4-point Likert scale (1 = never true, 4 = always true), covering communication (e.g., “My friends encourage me to talk about my difficulties”), trust (e.g., “I feel my friends are good friends”), and alienation (e.g., “I feel alone or apart when I am with my friends”). Higher scores on alienation were reverse-coded so that greater overall scores indicate stronger peer attachment. This scale was selected because of its well-documented validity in adolescent populations, including studies conducted among Canadian adolescents, where it demonstrated high internal consistency (Cronbach’s α = 0.80 in a past study, 0.82 in Wave 1 and 0.79 in Wave 2 in our sample) [47]. A composite score was generated by averaging the communication and trust subscales and subtracting the alienation subscale.

#### 2.2.3. Habitual Smartphone Use

To assess habitual smartphone use, the previously validated absent-minded subscale of the Smartphone Usage Questionnaire was used [48], which captures the automatic and unintentional aspects of smartphone behaviors. The subscale consists of seven items rated on a 7-point Likert scale, with sample items such as “How often do you check your phone while interacting with other people?” and “How often do you find yourself checking your phone without realizing why you did it?”. Higher scores indicate greater habitual smartphone use, and a composite score was created by averaging all item responses. This subscale was chosen because it specifically measures habitual rather than problematic use, aligning with previous research on Canadian adolescents where it showed strong internal reliability (Cronbach’s α = 0.91 in a past study, 0.87 in our sample) [49].

#### 2.2.4. Stress

Stress was examined using the Depression, Anxiety, and Stress Scale (DASS-21), a well-validated tool for assessing psychological distress in adolescents [50]. The stress subscale consists of seven items rated on a 4-point Likert scale (0 = did not apply, 3 = applied most of the time), with sample items including “I found it hard to wind down” and “I tended to overreact to situations”. Higher scores indicate greater levels of stress, and total scores were obtained by summing all item responses and multiplying by two, following the standard scoring procedure. This scale was selected because it has been widely used in adolescent mental health studies and has shown strong validity and reliability among Canadian youth (Cronbach’s α = 0.83 in a previous study, 0.85 in our sample) [51].

#### 2.2.5. Family Attachment

To assess family attachment, the same IPPA-SV that was used to determine peer attachment was used to investigate participants’ attachment with their families (twelve items), including three subsequent sections with four items each rated on a 4-point Likert scale: communication (e.g., “I tell my parents about my problems and troubles”), trust (e.g., “My parents accept me as I am”), and alienation (e.g., “I feel angry with my parents”). Items with higher scores indicating weaker family attachment were reverse-coded first, after which higher scores mean adolescents have stronger family attachment. The composite score was calculated by summing up communication and trust average scores and subtracting the average of alienation scores [52]. The IPPA-SV has been widely used in research on adolescent-parent relationships and has been validated in Canadian adolescent populations, showing strong internal consistency (Cronbach’s α = 0.82 in a past study, 0.87 in our sample) [53].

#### 2.2.6. School Connectedness

Participants completed the previously validated Perceived School Connectedness Questionnaire [54]. The scale contained nine items on a 5-point Likert scale with scores ranging from 1 (strongly disagree) to 5 (strongly agree). Sample items are “I feel close to people at school” and “I feel safe at school”. Items with higher scores indicating weaker school connectedness were reverse-coded first, after which higher scores indicate stronger school connectedness. The composite score was created by averaging the scores across all items. This scale was chosen because school connectedness is a key protective factor for adolescent well-being, and it has been widely validated in studies involving Canadian adolescents, demonstrating good internal reliability (Cronbach’s α = 0.80 in a past study, 0.75 in our sample) [55].

#### 2.2.7. Social Goals

Participants’ social goals were measured by two previously validated subscales: social popularity (six items, e.g., “Do you want to be popular among your peers?”) and social preference (four items, e.g., “Do you want to be accepted by your peers?”) [29]. All ten items are rated on a 5-point Likert scale ranging from 1 (never) to 5 (all of the time). Higher scores indicate an elevated level of social goals. Composite scores were obtained by calculating the average scores of each subscale. This measure was selected because it differentiates between popularity-seeking behaviors and the desire for peer acceptance, which are both crucial aspects of adolescent social adjustment. It has demonstrated strong validity in Canadian adolescent samples, with Cronbach’s α = 0.83 for social popularity and 0.73 for social preference in a past study [56] and Cronbach’s α = 0.88 for social popularity and 0.79 for social preference in our sample.

### 2.3. Statistical Analysis

Descriptive statistics, including mean and SD, and Pearson correlation coefficients of the main variables were presented. The number of peer attachment profiles in both waves was identified by conducting random intercept latent transition analysis (RI-LTA). A simulation study indicates that RI-LTA could have adequate statistical power to generate good results with two waves, a sample size of over 500, and continuous latent profile indicators [57]. RI-LTA separates between-subject variation from the within-subject transitions across waves. Thereby, the latent profile transitions across waves refer to within-subject transitions [58]. Due to the nature of RI-LTA, separate latent profile analysis at each wave is not a subset of joint RI-LTA across waves; the optimal number of profiles was chosen by fitting RI-LTA across waves jointly. Bayesian Information Criterion (BIC) was recommended to be used to determine the optimal number of profiles, with lower values suggesting better model fit [59,60]. Akaike Information Criterion (AIC) and adjusted BIC were further included to strengthen the rationale of selecting the number of profiles, with lower values suggesting better model fit [61]. Although entropy was not suggested to guide the decision of the number of profiles [62], we also reported them to indicate the profile classification accuracy. Following this, the stability and transition probabilities of peer attachment were examined by RI-LTA. Multivariable multinomial logistic regressions were then employed to assess the effect of habitual smartphone use exerted on the peer attachment profile transition probabilities, controlling for age, sex, ethnicity, stress, family attachment, school connectedness, and both social goals. A listwise deletion approach was taken where only those who reported complete data on all included variables were considered for analysis. Odds ratios with 95% confidence intervals were reported. All statistical analyses were executed on SAS version 9.4 and Mplus version 8.11.

## 3. Results

### 3.1. Descriptive Analysis

A total of 2446 adolescents participated in our survey at Wave 1, and 544 of them were excluded because they were either in grade 12 or categorized as “other sex”. In the end, 1902 high school students from Wave 1 were included in our analysis. Of the 1902 adolescents, 1303 of them joined both waves. Of the 1303 adolescents, the average age was 14.51 (SD 1.17), and 50.3% were female. Grades range from grade 8 to grade 12, and most of them were Asian (69.8% Asian and 18.6% White). Habitual smartphone use level was mid-range (mean = 4.02, SD = 1.09). The attrition rate is 31.49%, and the attrition analysis indicated that age (t = −5.89, *p* < 0.001; M_responded_ = 14.51, SD = 1.17, M_not-responded_ = 14.85, SD = 1.17), social popularity goals (t = −2.93, *p* = 0.004; M_responded_ = 2.46, SD = 0.86, M_not-responded_ = 2.59, SD = 0.85), social preference goals (t = 2.00, *p* = 0.046; M_responded_ = 3.83, SD = 0.85, M_not-responded_ = 3.74, SD = 0.92), and race (chi-square = 13.67, *p* = 0.001; %_White-responded_ = 63.02, %_Asian-responded_ = 71.26) were significantly different between the 2 groups. For all the variables in the 1303 participants, missing data rates ranged from 2.30% to 16.04%. Little’s Missing Completely at Random (MCAR) test indicated that the data were missing completely at random (χ^2^(1146) = 1195, *p* = 0.152).

### 3.2. Correlation Analysis

Correlation analyses showed that habitual smartphone use was positively associated with stress (r = 0.29, *p* < 0.001), social popularity goal (r = 0.20, *p* < 0.001), and social preference goal (r = 0.19, *p* < 0.001). Conversely, family attachment (r = −0.22, *p* < 0.001) and school connectedness (r = −0.18, *p* < 0.001) showed negative associations with habitual smartphone use. Negative associations were also revealed between habitual smartphone use and peer attachment at Wave 1 (r = −0.08, *p* < 0.01) (Table 1).

### 3.3. Profile Enumeration

Results indicated that the values of AIC, BIC, and aBIC decreased when the number of profiles increased from two to four (Table 2). A closer examination of the four-profile solution revealed that participants in the same profile did not have consistent patterns across peer attachment indicators, resulting in difficulties in defining profiles. Indeed, the number of profiles is decided according to model fit indices, interpretability, and theories [53]. As such, the three-profile solution best represents the current data at both waves and will be used for further analysis.

Profile 1, “Weak communication and some alienation” (Wave 1 prevalence: 39.00%), was characterized by having the lowest intercept values for all four items assessing communication among three profiles and medium-level intercept values for the four alienation items. Concerning trust, the intercept values for two items are the lowest, and another two items illustrate medium-level intercept values. Profile 2, “Strong communication, strong trust, and weak alienation” (Wave 1 prevalence: 51.05%), had the highest intercept values across all twelve items among the three profiles. Profile 3, “Okay communication and high alienation” (Wave 1 prevalence: 9.95%), had medium-level communication intercept values and the highest alienation intercept values across the three profiles. Similarly to Profile 1, two items for examining trust showed the lowest intercept values, and another two had medium-level intercept values among the three profiles (Figure 1).

### 3.4. RI-LTA

The multivariable RI-LTA model showed that 60.30%, 64.40%, and 33.20% of the participants remained stable at Profile 1, Profile 2, and Profile 3 from Wave 1 to Wave 2, respectively. With regard to profile transitioning, 31.30% and 4.30% transitioned from Profile 2 to Profile 1 and 3, respectively. In addition, 13.0% of the participants in Profile 1 at Wave 1 transitioned to Profile 3 at Wave 2 (Table 3).

Univariable RI-LTA showed that increased habitual smartphone use was associated with a heightened probability of transitioning from Profile 2 to Profile 1 (OR 1.28, 95% CI 1.08–1.51). This association persisted after only controlling for demographics (OR 1.33, 95% CI 1.12–1.58) and controlling for all covariates (OR 1.21, 95% CI 1.003–1.46). Moreover, participants having stronger social popularity goals had a decreased probability of transitioning from Profile 3 to Profile 2 (OR 0.58, 95% CI 0.41–0.83) (Table 4).

## 4. Discussion

In an increasingly digital world, adolescents are spending a significant amount of time on their smartphones, raising important questions about how habitual smartphone use impacts their social relationships. This study sought to answer the overarching research question: How does habitual smartphone use influence adolescents’ peer friendships over time? To investigate this, we used a novel approach to assess smartphone use behaviors—moving beyond conventional measures that focus on negative consequences. Instead, we employed the absent-minded smartphone use scale, a validated tool that objectively captures habitual smartphone behaviors and has been shown to outperform many existing assessment methods [49]. By applying this measure to a large cohort of Canadian adolescents, we found that higher levels of habitual smartphone use at baseline were associated with weaker peer attachment after one year, a pattern that persisted even after controlling for demographics and all other potential covariates. More specifically, adolescents with higher habitual smartphone use were 21% more likely to transition to a peer attachment profile characterized by reduced communication, lower trust, and increased alienation from their peers. Our findings provide empirical support for our three research objectives. Objective 1 aimed to assess whether habitual smartphone use predicts latent peer attachment profile transitions over time. The observed negative association is consistent with the displacement hypothesis, which suggests that excessive smartphone use may reduce opportunities for meaningful face-to-face interactions with peers [35]. Given the high prevalence of adolescent screen time in the digital era [63], this finding is not surprising.

Objective 2 examined whether demographic factors influenced this association, and results showed that the relationship remained stable even after accounting for age, sex, and other demographic variables. This aligns with previous research indicating that problematic smartphone use negatively affects peer attachment, regardless of demographic differences [4,38]. For example, cross-sectional evidence suggests that problematic smartphone use was negatively associated with peer attachment while controlling for sex [4]. In addition, longitudinal evidence further strengthened the findings that problematic smartphone use can lead to poorer peer attachment after controlling for age and sex [38].

Lastly, objective 3 tested the robustness of this association by controlling for all potential covariates. The persistence of this relationship after adjusting for multiple confounding factors suggests that habitual smartphone use is an independent predictor of weaker peer attachment. In unpacking these multivariable findings, we note that our results indicated that habitual smartphone use was negatively associated with school connectedness and family attachment, both of which could lead to strong peer attachment [20,24]. It is possible that when adolescents do not feel accepted by schools and have poor relationships with their parents, they use smartphones as an alternative attachment figure to obtain the sense of security that they do not receive from schools or parents [64,65]. Over time, this pattern of behavior may result in less attachment with peers. Additionally, our results also indicated that habitual smartphone use was positively associated with stress. This aligns with previous studies linking habitual smartphone use to mental health outcomes, including elevated levels of stress, anxiety, and depression [66]. Adolescents encountering higher levels of stress could be choosing smartphones as a coping mechanism instead of seeking connections from their peers. Indeed, previous studies have shown that habitual smartphone use is a reward-based behavior [7]. Compared with dedicating a large amount of time to socializing with peers, habitually checking smartphones is a fast way to receive pleasurable rewards, suggesting that this behavior is used as a means to cope with psychological tension [16], which replaces time that would be otherwise used to seek support from peers [35], and this further supports our findings for objective 1.

It is also possible that habitual smartphone use itself could have undermined peer attachment. One previous study has identified two types of smartphone use that are associated with habitual response patterns: process use and social use. While social use was not found to be associated with problematic behaviors, like smartphone addiction, process uses such as consuming media, browsing the internet, and playing games were associated with maladaptive outcomes [11]. Therefore, it could be the case that process use dominated the habitual smartphone use behaviors among adolescents in our study, which may have led to detrimental effects on peer attachment. Additionally, since habitual smartphone use can happen during face-to-face conversation (as captured by one of our items, which asked, “How often do you check your phone while interacting with other people?”; referred to as ‘phubbing’), it is possible that this aspect of habitual smartphone use (e.g., concentrating on one’s smartphone and ignoring the person who is physically present) could have led to poorer peer attachment [67]. This fits with prior work that has demonstrated that phubbing behaviors are associated with stronger feelings of ostracization between peers, decreased communication quality, and lower relationship satisfaction [68,69]. Future research should be conducted to explore this further.

Another significant finding of this study was that participants who had higher social popularity goals transitioned into lower peer attachment profiles over time. Although it is somewhat contradictory that wanting more popularity would lead to less peer attachment, it has been documented that prioritizing popularity can be associated with increased engagement in bullying behaviors [27,28]. Said differently, while bullying has been shown to be an effective way to maintain popularity and high social status [70], it likely has a negative impact on peer attachment. It is possible that this finding reflects adolescents who are engaging in anti-social behavior online in order to maintain their popularity status, which has in turn led to poorer peer attachment.

Our findings highlighted two practical implications targeting mitigating habitual smartphone use and improving peer attachment, respectively. First, digital literacy education could be useful for high school adolescents to reduce any potentially maladaptive habitual smartphone use (e.g., process use). Existing literature has shown that adolescents can benefit from such education to increase their self-control and smartphone self-efficacy, both of which are associated with positive outcomes [71]. Digital literacy training should incorporate skills and efforts to reduce habitual smartphone use, which has been shown useful in previous research [72,73]. Second, acknowledging the inevitable habitual smartphone use due to the high social need of adolescents [3], social-emotional learning (SEL) programs could be offered to help teens focus on in-person relationship skills even if they use smartphones habitually. Indeed, a comprehensive meta-analysis has found that universal school-based SEL can significantly improve peer attachment [74]. SEL focuses on improving peer attachment rather than mitigating habitual smartphone use. Digital literacy education integrated with SEL programs could provide maximum benefits for adolescents to build good relationships with peers.

Despite the strength of our study, which employed a longitudinal design with a large sample size and person-centered approach, several limitations should be noted. First, our sample predominantly consisted of Asian and White adolescents. This may limit the generalizability of our findings to adolescents of other ethnicities. Second, as is the nature of most self-report measures, habitual smartphone use assessment was based on self-report, which makes it subject to information bias. Third, a listwise deletion approach was used, and adolescents remaining in the analysis may not represent our original cohort, which could reduce statistical power and also lead to information bias.

## 5. Conclusions

Overall, our study aimed to answer the research question: How does habitual smartphone use influence adolescents’ friendships over time? Our findings indicated that adolescents who engaged in higher levels of habitual smartphone use were more likely to experience weaker peer attachment over one year, even after controlling for demographic and other confounding factors. This suggested that habitual smartphone use may play a role in shaping adolescent peer relationships, potentially reducing opportunities for meaningful social interactions.

While these results align with concerns that smartphones can negatively impact friendships, it is important to take a balanced perspective. Not all smartphone use is harmful—intentional and socially engaging digital interactions may strengthen friendships, whereas habitual and absent-minded use may lead to disengagement from in-person connections. Rather than viewing smartphones as inherently detrimental, interventions should focus on digital literacy and promoting mindful smartphone habits, helping adolescents use technology in ways that enhance, rather than replace, peer relationships.

Future research should further examine the bidirectional nature of this relationship, as it remains possible that weaker peer attachment may also lead to increased habitual smartphone use. Additionally, exploring specific types of habitual smartphone behaviors, such as passive media consumption versus social engagement, would provide deeper insight into their unique effects on adolescent social development. Longitudinal and experimental studies are needed to clarify causal pathways and inform intervention strategies.

By answering our research question, this study contributes to the growing understanding of how habitual smartphone use influences adolescent friendships. Our findings highlight the need for education programs, parental guidance, and school-based interventions that encourage balanced digital engagement while fostering strong peer relationships. As smartphone use continues to evolve, a nuanced approach—one that recognizes both the risks and benefits—will be essential for supporting adolescent social well-being in the digital age.

## Figures and Tables

**Figure 1 ijerph-22-00489-f001:**
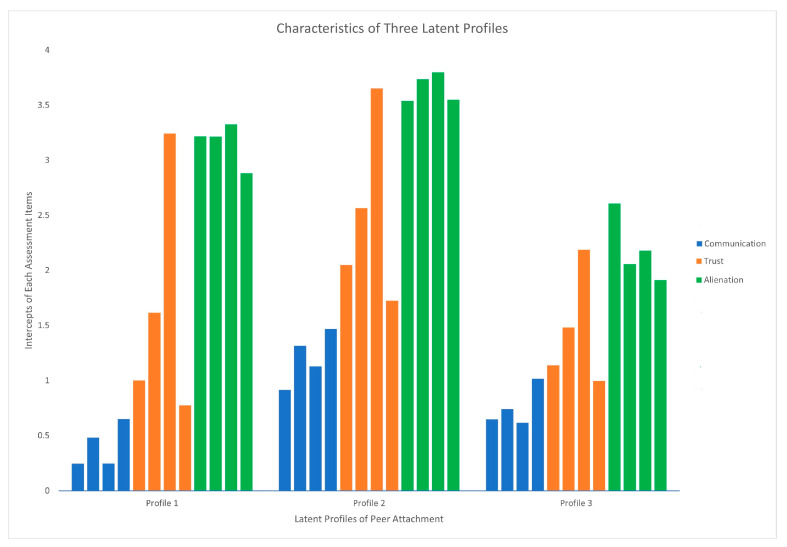
Profile intercepts of peer attachment for the three-profile model in RI-LTA.

**Table 1 ijerph-22-00489-t001:** Correlations between main variables.

	1	2	3	4	5	6	7	8
1 Habitual Smartphone Use	1							
2 Stress	0.29 ***	1						
3 Family Attachment	−0.22 ***	−0.31 ***	1					
4 School Connectedness	−0.18 ***	−0.28 ***	0.41 ***	1				
5 Social Popularity Goal	0.20 ***	0.13 ***	−0.04	0.01	1			
6 Social Preference Goal	0.19 ***	0.14 ***	0.03	0.10 ***	0.42 ***	1		
7 Peer Attachment Wave 1	−0.08 **	−0.16 ***	0.28 ***	0.39 ***	−0.15 ***	0.08 **	1	
8 Peer Attachment Wave 2	−0.05	−0.03	0.11 ***	0.19 ***	−0.09 **	0.09 **	0.44 ***	1
Mean	4.02	1.63	3.41	3.32	2.46	3.83	3.51	3.47
SD	1.09	1.05	1.74	0.61	0.86	0.85	1.51	1.41

** *p* < 0.01; *** *p* < 0.001; SD: standard deviation.

**Table 2 ijerph-22-00489-t002:** RI-LTA model fit indices.

Profiles	AIC	BIC	aBIC	Entropy
Univariable				
2	61,454.48	61,793.49	61,583.84	0.75
3	60,338.56	60,775.16	60,505.16	0.76
4	59,873.41	60,479.51	60,104.69	0.70
Controlling for demographics				
2	61,396.94	61,797.59	61,549.82	0.75
3	60,279.33	60,818.66	60,485.13	0.76
4	59,832.14	60,582.07	60,118.30	0.70
Controlling for all covariates				
2	58,059.37	58,530.47	58,235.07	0.77
3	57,003.54	57,662.07	57,249.14	0.78
4	56,485.52	57,402.39	56,827.47	0.73

AIC: Akaike Information Criterion; BIC: Bayesian Information Criterion; aBIC: Adjusted Bayesian Information Criterion.

**Table 3 ijerph-22-00489-t003:** Latent transition probabilities for three-profile model.

	Wave 2 Profiles		
Wave 1 Profiles	Profile 1	Profile 2	Profile 3
Profile 1	0.603	0.267	0.130
Profile 2	0.313	0.644	0.043
Profile 3	0.285	0.383	0.332

Profile 1: weak communication and some alienation; Profile 2: strong communication, strong trust, and weak alienation; Profile 3: okay communication and high alienation.

**Table 4 ijerph-22-00489-t004:** Multivariable RI-LTA of habitual smartphone use and all covariates on the latent profiles of peer attachment.

	Profile 2 vs. Profile 1			Profile 3 vs. Profile 1			Profile 3 vs. Profile 2		
Covariates	OR	LCL	UCL	OR	LCL	UCL	OR	LCL	UCL
Habitual Smartphone Use	**1.21**	1.003	1.46	0.97	0.73	1.29	0.80	0.60	1.07
Age	0.93	0.79	1.09	1.04	0.82	1.31	1.12	0.90	1.41
Female (vs. Male)	0.80	0.53	1.19	1.15	0.63	2.09	1.44	0.81	2.57
Asian (vs. White)	0.83	0.51	1.35	1.35	0.70	2.61	1.63	0.84	3.17
Other Races (vs. White)	1.16	0.53	2.51	1.01	0.39	2.64	0.87	0.35	2.16
Stress	1.06	0.86	1.30	1.17	0.85	1.61	1.11	0.80	1.52
Family Attachment	0.94	0.83	1.07	1.04	0.89	1.23	1.11	0.94	1.31
School Connectedness	0.78	0.54	1.13	1.03	0.62	1.69	1.31	0.78	2.20
Social Popularity Goal	1.25	0.97	1.60	0.73	0.51	1.03	**0.58**	0.41	0.83
Social Preference Goal	0.95	0.74	1.23	1.25	0.84	1.87	1.31	0.88	1.96

Bolded OR represents *p* < 0.05. Profile 1: weak communication and some alienation; Profile 2: strong communication, strong trust, and weak alienation; Profile 3: okay communication and high alienation.

## Data Availability

The data presented in this study are available on request from the corresponding author due to privacy or ethical restrictions.

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
