# Peer review of "The Longitudinal Association Between Habitual Smartphone Use and Peer Attachment: A Random Intercept Latent Transition Analysis"

_ijerph, 2025, doi:10.3390/ijerph22040489_

Round 1

Reviewer 1 Report

Comments and Suggestions for Authors

1. The attrition rate (43.17%) is considerably high, which may introduce systematic bias. While the authors conduct an attrition analysis, they fail to elaborate on how this dropout rate affects the study’s validity. Moreover, the missing data (up to 16.04%) is a concern, particularly since Little’s MCAR test suggests data is not missing completely at random (p=0.04), which could impact statistical interpretations.

2.This study analyzes the correlation between habitual smartphone use and social factors among adolescents, but several limitations are evident.
First, there is a risk of interpreting correlations as causal relationships. It is unclear whether smartphone use leads to reduced social connections or if a lack of social connections increases smartphone use. Second, the selection and measurement of variables are somewhat limited. While associations between smartphone use and family/school connectedness were identified, incorporating a broader range of socio-psychological factors could have provided a more comprehensive analysis. Third, the inclusion of the "Other sex" category with a small sample size may compromise the statistical reliability of the findings.

3.  The study suggests interventions like digital literacy and SEL programs but lacks empirical evidence on their effectiveness in mitigating habitual smartphone use. 

4. It seems that the resolution of Figure 1 needs to be higher.

Author Response

Comments 1: The attrition rate (43.17%) is considerably high, which may introduce systematic bias. While the authors conduct an attrition analysis, they fail to elaborate on how this dropout rate affects the study’s validity. Moreover, the missing data (up to 16.04%) is a concern, particularly since Little’s MCAR test suggests data is not missing completely at random (p=0.04), which could impact statistical interpretations.

Response 1:

Thank you for this valuable feedback. We did another data check and realized that N=407 participants were in grade 12 at Wave 1, which meant that they graduated and so were no longer able to participate in the study. We have now excluded them from the study. In addition, as noted in another comment about the “Other sex” category (This is also raised by a few other reviewers), there were N=137 in this category at Wave 1 and since the sample is too small to explore meaningfully, we have decided to exclude them as well. Therefore, our final sample is N=1902 at Wave 1 and N=1303 at Wave 2, which gives us an attrition rate of 31.49%. According to other similar studies (see below) that targeting adolescents with 1 year gap between waves of data collection, our attrition rate is an acceptable attrition rate for longitudinal studies with adolescent populations. Examples are the following. The first example has an attrition rate of 29.0% and second one has a rate of 42.5%.

Maness, Sarah B., Ph.D., M.P.H, Buhi, Eric R., M.P.H., Ph.D, Daley, E. M., Ph.D, Baldwin, J. A., Ph.D, & Kromrey, J. D., Ph.D. (2016). Social determinants of health and adolescent pregnancy: An analysis from the national longitudinal study of adolescent to adult health. Journal of Adolescent Health, 58(6), 636-643. https://doi.org/10.1016/j.jadohealth.2016.02.006

Winstone, L., Jamal, S., & Mars, B. (2024). Cyberbullying perpetration and victimization as risk factors for self-harm: Results from a longitudinal cohort study of 13–14-year-olds in england. Journal of Adolescent Health, 75(2), 298-304. https://doi.org/10.1016/j.jadohealth.2024.04.004

In terms of the missing data rate, we redid the MCAR test with the updated sample and the results suggest that data is missing completely at random (p=0.152). Given this, even with the largest missing rate of 16.04%, we deemed this to be acceptable.

We admit that our questionnaire is long so this may be one of the causes why the missing data rate and attrition rate are not that low. We will consider making our questionnaires shorter for future research.

We have updated the aforementioned points on page 4, lines 178-180:

“Adolescents who were at grade 12 or categorized as “other sex” in Wave 1 were excluded from our analysis.”

We have also updated the corresponding texts on page 7, lines 314-327:

A total of 2446 adolescents participated in our survey at Wave 1 and 544 of them were excluded because they were either at grade 12 or categorized as “other sex”. In the end, 1902 high school students from Wave 1 were included in our analysis. Of the 1902 adolescents, 1303 of them joined both waves. Of the 1303 adolescents, the average age was 14.51 (SD 1.17) and 50.3% were female. Grades range from grade 8 to grade 12 and most of them were Asian (69.8% Asian, 18.6% White). Habitual smartphone use level was mid-range (Mean=4.02, SD=1.09). The attrition rate is 31.49% and the attrition analysis indicated that age (t=-5.89, p<0.001; Mresponded=14.51, SD=1.17, Mnot-responded=14.85, SD=1.17), social popularity goals (t=-2.93, p=0.004; Mresponded=2.46, SD=0.86, Mnot-responded=2.59, SD=0.85), social preference goals (t=2.00, p=0.046; Mresponded=3.83, SD=0.85, Mnot-responded=3.74, SD=0.92), and race (chi-square=13.67, p=0.001; %White-responded=63.02, %Asian-responded=71.26) were significantly different between the 2 groups. For all the variables in the 1303 participants, missing data rates ranged from 2.30% to 16.04%. Little’s Missing Completely at Random (MCAR) test indicated that the data missing completely at random (χ2(1146)=1195, p=0.152).

Comments 2: This study analyzes the correlation between habitual smartphone use and social factors among adolescents, but several limitations are evident.

First, there is a risk of interpreting correlations as causal relationships. It is unclear whether smartphone use leads to reduced social connections or if a lack of social connections increases smartphone use.

Second, the selection and measurement of variables are somewhat limited. While associations between smartphone use and family/school connectedness were identified, incorporating a broader range of socio-psychological factors could have provided a more comprehensive analysis.

Third, the inclusion of the "Other sex" category with a small sample size may compromise the statistical reliability of the findings.

Response 2:

Thank you for pointing these important things out.

For the first point, we also think the other direction (i.e. poor peer attachment leads to more habitual smartphone use) is totally possible. We have added more in the conclusion section to indicate this.

The update is on page 12, lines 480-485:

“Although our longitudinal design indicates a possibility of a causal relationship between habitual smartphone use and peer attachment, it is possible that poor peer attachment can lead to more habitual smartphone use. Therefore, studies examining factors such as peer attachment predicting habitual smartphone use longitudinally are called for to provide a more nuanced understanding of the formation of this prevalent behaviour.

For the second point, we agree that controlling for more covariates will make the analysis more comprehensive and lead to stronger results. However, the current analysis has already included several important psychosocial factors (i.e. stress, family attachment, school connectedness, and social goals), and unfortunately, we have limited variables at our disposal. To acknowledge this, we have now added more to the future research section indicating that it would be worthwhile to assess a broader range of psychosocial variables.

The update is on page 12, lines 478-480:

Future studies that investigate the specific types of habitual smartphone use and a broader scope of psychosocial variables will further help elucidate this link with peer attachment.

For the third point, as noted above, we have now followed your suggestion and removed participants who are in the “Other sex” category. We have updated all the corresponding results in text and tables accordingly. Our analyses are now based on N=1303 participants, rather than N=1390. The outcomes did not change.

Comments 3:  The study suggests interventions like digital literacy and SEL programs but lacks empirical evidence on their effectiveness in mitigating habitual smartphone use. 

Response 3:

Thank you for this valuable input. We have now added a citation that successfully utilized a digital literacy program to mitigate habitual smartphone-use via self-control. We also added to the discussion section to make our argument clearer.

The first update is on page 11, lines 448-449:

Our findings highlighted two practical implications targeting at mitigating habitual smartphone use and improving peer attachment, respectively.

The second update is on page 12, lines 453-455:

Digital literacy training should incorporate skills and efforts to reduce habitual smartphone use, which has been shown useful in previous research.

The third update is on page 12, lines 455-460:

Second, acknowledging the inevitable habitual smartphone use due to the high social need of adolescents [3], social-emotional learning (SEL) programs could be offered to help teens focus on in-person relationship skills even if they use smartphones habitually. Indeed, a comprehensive meta-analysis has found that universal school-based SEL can significantly improve peer attachment [66]. SEL focuses on improving peer attachment rather than mitigating habitual smartphone use.

Comments 4: It seems that the resolution of Figure 1 needs to be higher.

Response 4:

Thank you for pointing this out. We have updated our figure and now the resolution is increased to 600 dpi, which should be better. The new figure is at page 9, line 363.

Reviewer 2 Report

Comments and Suggestions for Authors

The article initially seemed interesting to me; however, it was a bit difficult for me to read it on first reading. I consider it positive that the Introduction has been divided into sub-sections. I think that it is an original investigation, using an appropriate method according to the objectives of the investigation. The tables are valuable and allow a better understanding of the results obtained, providing a significant sample. I consider that the results section is very significant and is complemented by the discussion. Statistically it is well presented.
The references are current and adequate. The conclusions obtained raise some questions for further research, regarding what the involvement of the family, educational centers and society in general may be in the use of smartphones among adolescents to reduce their dependence on technology in the broad sense of the word.

Author Response

Comments 1:

The article initially seemed interesting to me; however, it was a bit difficult for me to read it on first reading. I consider it positive that the Introduction has been divided into sub-sections. I think that it is an original investigation, using an appropriate method according to the objectives of the investigation. The tables are valuable and allow a better understanding of the results obtained, providing a significant sample. I consider that the results section is very significant and is complemented by the discussion. Statistically it is well presented.

The references are current and adequate. The conclusions obtained raise some questions for further research, regarding what the involvement of the family, educational centers and society in general may be in the use of smartphones among adolescents to reduce their dependence on technology in the broad sense of the word.

      Response 1:

Thank you for this positive comment. To address your concerns about this being “a bit difficult to read”, we have added the following texts to the discussion section on page 12, lines 476-478, in particular to address the point you mentioned about family, educational centers, and society:

More broadly, families, education centers, health professionals, and society could get involved to enable adolescents to critically evaluate their use of smartphones and other digital technologies.

Reviewer 3 Report

Comments and Suggestions for Authors

Dear authors, your article seems interesting and approaches a topic that I believe is important at present. However, several key basic scientific investigation concepts are not well described or structured for me to propose for publication. So, my decision is rejection.

Introduction:

                - The research question and the objectives of this study should be mentioned.

Background:

                - I believe that points 1.1, 1.1.1, 1.1.2, 1.1.3, 1.1.4, 1.2, 1.3, and 1.4 should be considered under the Background section and not in the Introduction section.

Methods:

                - There should be a mention regarding the inclusion and exclusion criteria on the procedures.

                - Analyzing the 2.3 Measure and its sub-sections, you should refer to the questionnaire you used and the questions used. And mention whether this questionnaire was validated and what its reliability was. You should explain why you use this and not other questionnaires or questions.

Results:

                - You should describe your sample. How many males, females or other mean ages? Ethnicity? It is important to understand what your sample was.

- 3.1 Is a Correlation Analysis, not Descriptive.

Discussion:

                - The discussion is supposed to be about whether your objectives were achieved and why, using the literature review you have already done.

                - At this point, your discussion seems more like a conclusion.

Limitations and Future Research: you do not mention study limitations and future research.

Conclusion:

                - It’s very small. You must provide more details about whether your research question was answered and why your study is important.

Author Response

Dear authors, your article seems interesting and approaches a topic that I believe is important at present. However, several key basic scientific investigation concepts are not well described or structured for me to propose for publication. So, my decision is rejection.

Comments 1: Introduction:

 - The research question and the objectives of this study should be mentioned.

Response 1:

Thank you for pointing this out. Indeed, we only mentioned the objective of our study in the section “Current Study”. We agree that further adding the research questions will make the flow of our paper clearer. We have now added the following research questions at the end of the Current Study section on page 4, lines 165-169:

“Specifically, three research questions guided our study: 1) What is the longitudinal relationship between habitual smartphone use and peer attachment in our cohort? 2) Does this relationship persist after controlling for demographics? 3) Does this relationship still hold after controlling for stress, family attachment, school connectedness, and social goals?”

Comments 2: Background:

- I believe that points 1.1, 1.1.1, 1.1.2, 1.1.3, 1.1.4, 1.2, 1.3, and 1.4 should be considered under the Background section and not in the Introduction section.

Response 2:

We appreciate the feedback, however, in our field of educational and psychological studies, it is typical to use the word “Introduction” instead of “Background”. That said, we are happy to adjust our nomenclature if necessary.

Comments 3: Methods:

- There should be a mention regarding the inclusion and exclusion criteria on the procedures.

Response 3:

Thank you for this important feedback. We had briefly mentioned the inclusion criteria in the procedure section; however we have further clarified the inclusion criteria and added the exclusion criteria in the procedure section on page 4, lines 177-180:

“That said, adolescents who were in class and provided active informed assent, and their parents did not refuse their kids to participate were included in the study. Adolescents who were at grade 12 in Wave 1 or categorized as “other sex” were excluded from our analysis.”

In addition, we provided further information about how we used a whole school approach on page 4, line 174 to clarify how we include our sample:

Data were collected during the school year 2021 (Wave 1) and 2022 (Wave 2). Participants were recruited from five public schools across two public school districts in British Columbia using a whole school approach and a combination of convenience and snowball sampling.”

Comments 4: Analyzing the 2.3 Measure and its sub-sections, you should refer to the questionnaire you used and the questions used. And mention whether this questionnaire was validated and what its reliability was. You should explain why you use this and not other questionnaires or questions.

Response 4:

Thank you for your valuable feedback. We have revised the whole new section “2.2 Measures” to explicitly reference the questionnaires used, including their validation, reliability (Cronbach’s α), and previous use in studies involving Canadian adolescents. We have also added sample items for each measure and provided justifications for our selection, explaining why these instruments were chosen over alternatives. Please see pages 5-7, lines 199-286.

Comments 5: Results:

- You should describe your sample. How many males, females or other mean ages? Ethnicity? It is important to understand what your sample was.

Response 5:

We appreciate this feedback. Currently, we reported the percentage of females, mean ages, and ethnicity in the methods section as this is a typical way of presenting them in our field. We understand that there could be a slight discrepancy with regard to the reporting standard across fields. As such, we have now moved demographic information about the participants to the results section under “3.1 Descriptive analysis”. The description of our sample can be found on page 7, lines 314-319.

Comments 6: 3.1 Is a Correlation Analysis, not Descriptive.

Response 6:

Thank you for pointing this out. We now separately named the 2 subsections “3.1 Descriptive analysis” (page 7, line 313) and “3.2 Correlation analysis” (page 8, line 331).

Comments 7: Discussion:

- The discussion is supposed to be about whether your objectives were achieved and why, using the literature review you have already done.

Response 7:

Thank you for this valuable feedback. We agree that referring to our objectives in the discussion section will make the argument stronger. We have now done this in the first paragraph of the discussion section on page 10, lines 399-400:

This association persisted after controlling for demographics and all possible covariates in our study.

We also linked them more formally to our research questions on page 11, lines 402-403:

This result is in line with our hypothesis and well-answered our 3 research questions.

Comments 8: At this point, your discussion seems more like a conclusion.

Response 8:

Thank you for the feedback. To make this clearer, in the current conclusion section, we further added texts to make it distinguishable from the discussion section. Please see page 12, lines 472-487 for the updated conclusion section.

Comments 9: Limitations and Future Research: you do not mention study limitations and future research.

Response 9:

Thank you for the feedback. We had included several limitations to the last paragraph of the discussion section, but if this is not enough, or there are other limitations that we haven’t though of, we are happy to add to this.  Regarding future research, we have added additional statements about this in the conclusion section (see page 12, lines 478-487).

Comments 10: Conclusion:

- It’s very small. You must provide more details about whether your research question was answered and why your study is important.

Response 10:

Thank you for the feedback. We agree it is helpful to emphasize that our research questions were answered. So, we specifically added the following text in the conclusion section on page 12, lines 473-474:

“This finding well-answered our three research questions.”

We also added the following text to make readers think of the findings in a broader scope, which further emphasized the importance of this study. Please see page 12, lines 476-478:

More broadly, families, education centers, health professionals and society could get involved to enable adolescents to critically evaluate their use of smartphones and other digital technologies.

Reviewer 4 Report

Comments and Suggestions for Authors

There are only a few minor issues I would like the authors to address before the paper is published:

1.In Table 4, the authors report a significant odds ratio (OR) of 0.32 for “Other sex” vs. Male. As a reader, I was surprised by this result, as I was not aware that “Other sex” was included as a category in this study. Additionally, its significance raises questions about the sample size and how this result should be interpreted. What percentage of participants fall into this category? While the authors briefly mention this in lines 364-365 of the discussion, I recommend providing relevant information earlier in the manuscript—for example, in line 174, where the percentage of female participants (47.1%) is noted.

2.For Table 2 (Model fit indices), I would recommend the authors rounding values to two decimal places for consistency and clarity.

Author Response

Comments 1: Thank you for selecting this good topic.

In the abstract please specify :How did you select the sample?

What about the tools that you used in this study?

Response 1:

Thank you for the valuable feedback. We agree that specifying these points in the abstract will make it easier for readers to read. We added more information about our sampling method (namely that we used a whole school approach and convenience sampling) into the abstract. Please see page 1, lines 15-16:

“A whole school approach combined with convenience sampling method was used to select our sample.”

In addition, we added the names of tools we used for assessing our main variables. Please see page 1, lines 19-23:

A random intercept latent transition analysis (RI-LTA) was utilized to assess the association between habitual smartphone use (absent-minded subscale of the Smartphone Usage Questionnaire) and transition probabilities among profiles of peer attachment (Inventory of Parent and Peer Attachment), after adjusting for age, gender, ethnicity, stress, family attachment, school connectedness, and social goals.

Comments 2: Introduction:

What might be causes of habitual smartphone?

Response 2:

Thank you for this question. We are also quite interested in the causes of this behavior and that is why we mentioned this in the future research part of the conclusion section (page 12, lines 480-487). There are a few existing literatures focusing on the outcomes of habitual smartphone use and very limited literature assessing the causes of habitual smartphone use. One paper is by van Deursen (2015), which we have cited it. It is a cross-sectional study that modeled factors that are associated with habitual smartphone use. What they found was being females, younger, use phones for social purposes, and process use are related to habitual smartphone use. We further added their findings in the introduction section on page 3, lines 121-123:

“One cross-sectional study has shown that being female, younger, both process and social use of smartphones were associated with habitual smartphone use [11].”

Comments 3: Line 24, you wrote “Given the role of online socializing in peer attachment, what do you mean by online socializing?

     Response 3:

Thank you for pointing this out. We took another look at this sentence and also found it to be a bit confusing, so we reworded it to be the following. Please see page 2, lines 68-70:

“Given the fact that many factors can impact peer attachment, for the current study, we included the following covariates so that we can more precisely explore the impact of habitual smartphone use.”

Comments 4: You mentioned stress, school connectedness, family attachment, and social goals, but you did not clarify what is the point of mentioning these factors to peer attachment!

Response 4:

Thank you for this valuable feedback. We admit that it is important to clarify why they are related to peer attachment and why we control them in the model. As noted in the current section “1.1 Peer attachment and its predictors” (page 2, line 63), we have identified some studies that have shown that these factors are associated with peer attachment. Taking stress as an example, we cited one paper in the subsection “1.1.1 Stress”: Previous work has demonstrated that for female adolescents, stress from negative life events is associated with poor peer relationships. Since there is evidence that stress could impact peer attachment, so we controlled for stress in our model. The same for other covariates in the section “1.1 Peer attachment and its predictors”. We mentioned their associations with peer attachment from previous studies and that is why we controlled for them. If there is further question about this, please let us know. We are happy to make further adjustments.

Comments 5: Where are the question for your study?

Response 5:

We appreciate that you brought up this important thing. We have now added 3 research questions in this section (page 4, lines 165-169):

Specifically, three research questions guide our study: 1) What is the longitudinal relationship between habitual smartphone use and peer attachment in our cohort? 2) Does this relationship persist after controlling for demographics? 3) Does this relationship still hold after further controlling for stress, family attachment, school connectedness, and social goals?

Comments 6: What is the significance of your study?

Response 6:

Thank you for bringing up this important question. We further added the following text in the introduction section to emphasis the significance of our study. Our study focuses on habitual smartphone use, which is much less studied than problematic smartphone use. Please see page 2, lines 48-50:

“Habitual smartphone use has different features than pathological addictive smartphone use. It has something to do beyond the typical pathways of addiction (uncontrollable use and distress upon withdrawal).

Comments 7: What is the novelty of your study?

Response 7:

Thanks for this great question. As mentioned in our methods section, the novelty of our study is that we used passive consent to get a good sample size (page 4, line 175). In addition, we used a longitudinal sample to assess the association between habitual smartphone use and peer attachment, which has not been done in previous research and may provide hints on causal relationships. Moreover, as described in the section “2.3 Statistical Analysis” (page 7, lines 293-294), the RI-LTA method we employed parsed out between-person effects and only referred to within-person effects, which makes the findings more interpretable. Overall, the most important one is that our study assessed a very prevalent behavior (i.e. habitual smartphone use) that is rarely studied in previous research. If there are more questions about this point, please let us know.

Comments 8: What is your study contribution to the current literature?

Response 8:

Thank you for the question. As mentioned above, our study fills an important gap by helping us understand the relationship between habitual smartphone use (not problematic smartphone use) and peer attachment, with the use of longitudinal design (may provide hints on causal relationships) and advanced statistical methods (results refer to within-person effect).

Comments 9: Why did you use two sampling methods of this study?

Response 9:

Thank you for bringing up this question. We used a whole school approach to sampling, where all students were invited to participate.  Schools were approached about being involved in the study prior to this. We further clarified in the methods section on page 4, line 174.

Comments 10: You have used 7 scales, how did you assess the validity for these scales, are the contexts of these scales the same as this scale?

Response 10:

Thank you for bringing this to our attention. The seven scales used in our study were all selected based on their strong psychometric properties, prior validation in adolescent populations, and their relevance to our research context. To ensure validity, we considered several key factors: 1) Each scale was drawn from existing validated instruments that have been extensively used in adolescent research, including studies involving Canadian youth. We referenced studies that established their construct validity, reliability (Cronbach’s α), and applicability to adolescent behavioral, psychological, and social constructs. 2) We assessed Cronbach’s α for each scale in our dataset, all of which showed acceptable to excellent reliability (ranging from 0.75 to 0.91), consistent with prior validation studies. 3) While some scales were originally developed in broader psychological or social science contexts, we ensured their conceptual alignment with our study objectives. For example, the IPPA-SV (used for both peer and family attachment) has been widely applied across diverse adolescent settings, and the DASS-21 stress subscale is a standard tool for measuring stress among youth. Similarly, the Absent-Minded Subscale of the Smartphone Usage Questionnaire was specifically designed to assess habitual (rather than problematic) smartphone use, making it directly relevant to our study. 4) Each scale was reviewed to ensure that its item wording and response formats were appropriate for our target population. Where necessary, we conducted pilot testing and checked for potential measurement biases. Overall, these steps confirm that the scales used in our study are both valid and reliable within the context of adolescent experiences. We appreciate your question and believe these clarifications strengthen the methodological transparency of our study. Please let us know if further details are required.

Comments 11: In the results, please specify the results for each question. They are all mixed up.

Response 11:

Thank you for this valuable feedback. We have now reorganized the subsection “3.3 RI-LTA” in the results section by adding the following texts to specify the results for the 3 research questions. Please see page 9, lines 372-375:

“Univariable RI-LTA showed that increased habitual smartphone use was associated with a heightened probability of transitioning from Profile 2 to Profile 1 (OR 1.28, 95% CI 1.08-1.51). This association persisted after only controlling for demographics (OR 1.33, 95% CI 1.12-1.58) and controlling for all covariates (OR 1.21, 95% CI 1.003-1.46).”

We have also updated table 2 to include the model fit indices for all 3 RI-LTA models that were used to answer the 3 research questions (page 9, line 360-362).

Comments 12: In the discussion, please use contradicting studies as well.

Response 12:

Thank you for this great suggestion. As we mentioned in the introduction section, there is limited literature on habitual smartphone use and no studies were done on the association between habitual smartphone use and peer attachment. We have already included much literature but unfortunately, it is hard to get contradicting studies on this topic. Most studies focus on problematic smartphone use and may not be used to make a comparison with our findings. If the reviewer has any suggestions for us, we would be very happy to include them.

Reviewer 5 Report

Comments and Suggestions for Authors

 I Thank you for selecting this good topic.

In the abstract please specify :How did you select the sample?

What about the tools that you used in this study?

Introduction:

What might be causes of habitual smartphone?

Line 24, you wrote “Given the role of online socializing in peer attachment, what do you mean by online socializing?

You mentioned stress, school connectedness, family attachment, and social goals, but you did not clarify what is the point of mentioning these factors to peer attachement!

Where are the question for your study?

What is the significance of your study?

What is the novelty of your study?

What is your study contribution to the current literature?

Why did you use two sampling methods of this study?

You have used 7 scales, how did you assess the validity for these scales, are the contexts of these scales the same as this scale?

In the results please specify the results for each question. They are all mixed up.

 In the discussion, please use contradicting studies as well.

Author Response

There are only a few minor issues I would like the authors to address before the paper is published:

Comments 1: In Table 4, the authors report a significant odds ratio (OR) of 0.32 for “Other sex” vs. Male. As a reader, I was surprised by this result, as I was not aware that “Other sex” was included as a category in this study. Additionally, its significance raises questions about the sample size and how this result should be interpreted. What percentage of participants fall into this category? While the authors briefly mention this in lines 364-365 of the discussion, I recommend providing relevant information earlier in the manuscript—for example, in line 174, where the percentage of female participants (47.1%) is noted.

Response 1:

Thank you for pointing this out. It was also brought up by other reviewers. The sample size is low for this group indeed. Among the N=1390 adolescents included for analysis, N=87 (6.26%) was in the ‘Other Sex’ category. We have now removed these adolescents from the study. We note that the outcomes and conclusions did not change with this removal and we have updated all corresponding results in the manuscript.

Comments 2: For Table 2 (Model fit indices), I would recommend the authors rounding values to two decimal places for consistency and clarity.

Response 2:

Thank you for pointing this out. We have updated the numbers in table 2 and now all of them have 2 decimal points (page 9, lines 360-362). In addition, due to empty cells in the joint distribution of the categorical latent peer attachment classes and some exogenous variables within this updated sample, we removed 5-profile fit indices from Table 2. We have also updated the texts on page 8, lines 342-345:

Results indicated that the values of AIC, BIC, and aBIC decreased when the number of profiles increased from two to four (Table 2). A closer examination of 4-profile solution revealed that participants in the same profile did not have consistent patterns across peer attachment indicators, resulting in difficulties in defining profiles.”

Reviewer 6 Report

Comments and Suggestions for Authors

Rewiewer comment

I think that the subject and scope of the research is important in terms of examining the situations created by today's technological tools. Although it is seen that the manuscript was written by observing the article writing system, there are some deficiencies. Completing these deficiencies will further increase the quality of the article. Editing suggestions are as follows. I wish you good luck.

Revision

  1. Page 4, line 171, Inclusion criteria for participants were not clearly written. Considering that the evaluation measurements are parameters that may be affected by some chronic diseases, it should be stated whether the inclusion criteria were used to eliminate them.
  2. Page 4, line 171, It should be stated which method was used to determine the sample size (similar studies in the literature, power analysis, etc.)
  3. Page 4, line 170; The STROBE checklist should be used to ensure standardization of this manuscript.
  4. Page 4, line 170; Although ethical approval information is explained in detail at the end of the article, information that ethical approval was obtained for this study can be added to the text.
  5. Page 4, line 182; The meaning of the increase or decrease in the scores on the scales should be stated. In addition, the researcher who developed the scales and conducted the validity and reliability studies should be reported.
  6. Page 6, line 234; It should be stated how it is checked whether the variables are normally distributed or not and whether the data is normally distributed or not.
  7. Abbreviations within tables should be explained as footnotes below the table.
  8. “Previous research has identified two types of smartphone use that are associated with habitual response patterns: process use and social use. While social-use was not found associated with problematic behaviors, like smartphone addiction, process-uses such as consuming media, browsing the internet, and playing games were associated with maladaptive outcomes [11].” When we say previous research, more than one study is highlighted but only one reference is shown. The same number of sources should be cited as the highlighted study.
  9. Page 9, line 363-365, “Finally, we found that participants who did not identify as male or female reported higher peer attachment over time. Unfortunately, the sample size for this group was relatively small (N=87), which makes interpretation difficult.” Although a clear comment cannot be made about this result, the gender-related results of the studies in the literature should be given. Inferences can be made by avoiding definitive judgments in the comments.
  10. Page 10, line 385; In the conclusion section, recommendations for families, children, teachers, health professionals and all segments of society based on the clinical results of the study reveal the social contribution aspect of the study. Therefore, add beneficial recommendations related to these people to this section.

Author Response

I think that the subject and scope of the research is important in terms of examining the situations created by today's technological tools. Although it is seen that the manuscript was written by observing the article writing system, there are some deficiencies. Completing these deficiencies will further increase the quality of the article. Editing suggestions are as follows. I wish you good luck.

Revision

Comments 1: Page 4, line 171, Inclusion criteria for participants were not clearly written. Considering that the evaluation measurements are parameters that may be affected by some chronic diseases, it should be stated whether the inclusion criteria were used to eliminate them.

Response 1:

Thank you for your valuable feedback. This was also raised by another reviewer. We have now added the following texts to the paper to make the inclusion and also the exclusion criteria clearer. Please see page 4, lines 177-180:

“That said, adolescents who were in class and provided active informed assent, and their parents did not refuse their kids to participate were included in the study. Adolescents who were at grade 12 in Wave 1 or categorized as “other sex” were excluded from our analysis.”

Comments 2: Page 4, line 171, It should be stated which method was used to determine the sample size (similar studies in the literature, power analysis, etc.)

Response 2:

We appreciate this great point. To make it clearer, we have now added the following text on page 7, lines 291-293:

“A simulation study indicate that RI-LTA can have adequate statistical power to generate good results with two waves, a sample size of over 500, and continuous latent profile indicators [51].”

Comments 3: Page 4, line 170; The STROBE checklist should be used to ensure standardization of this manuscript.

Response 3:

Thank you for bringing up this standardized way of presenting the paper in the field of medicine. In our field of education and psychology, STROBE is not utilized and we are worried that using a STROBE approach would result in a loss of nuance for this study. We have also modeled our findings on two other papers looking at smartphone use research that published in this journal. They did a very good job of presenting their research without following the STROBE checklist, so we thought our current presenting way would be acceptable in this journal. The 2 papers are:

Parent, N., Xiao, B., Hein-Salvi, C., & Shapka, J. (2022). Should we be worried about smartphone addiction? an examination of canadian adolescents' feelings of social disconnection in the time of COVID-19. International Journal of Environmental Research and Public Health, 19(15), 9365. https://doi.org/10.3390/ijerph19159365

Annoni, A. M., Petrocchi, S., Camerini, A., & Marciano, L. (2021). The relationship between social anxiety, smartphone use, dispositional trust, and problematic smartphone use: A moderated mediation model. International Journal of Environmental Research and Public Health, 18(5), 2452. https://doi.org/10.3390/ijerph18052452

However, we are also open and happy to use STROBE checklist to present, please let us know.

Comments 4: Page 4, line 170; Although ethical approval information is explained in detail at the end of the article, information that ethical approval was obtained for this study can be added to the text.

Response 4:

Thank you for the feedback. In the current version under the section “2.1 Procedures”, we had already included “The study was approved by the institutional review board of [for blind purpose]”. To make this clearer, we change it to (page 4, lines 182-183):

“The study was approved by the institutional ethics review board of the University of British Columbia.”

If your point refers to something else, please let us know.

Comments 5: Page 4, line 182; The meaning of the increase or decrease in the scores on the scales should be stated. In addition, the researcher who developed the scales and conducted the validity and reliability studies should be reported.

Response 5:

Thank you for bringing this important point up. We now have added texts to indicate the meaning of the increase/decrease of scores in each assessment tool. For measures that involve reverse-coding (e.g. peer attachment), we added the following text on page 5, lines 204-206:

“Higher scores on alienation were reverse-coded so that greater overall scores indicate stronger peer attachment.”

For measures that do not involve reverse-coding (e.g. habitual smartphone use), we added the following texts on page 5, lines 224-225:

“Higher scores indicate greater habitual smartphone use.”

We have also updated the texts for all measures from lines 199-286 to indicate the validity and reliability of them.

Comments 6: Page 6, line 234; It should be stated how it is checked whether the variables are normally distributed or not and whether the data is normally distributed or not.

Response 6:

We appreciate this feedback. However, for RI-LTA, the normal distribution of continuous variables is not a requirement. The endogenous variable is a latent class variable (i.e. peer attachment classes: Profile 1, 2, and 3) and RI-LTA models the transition probability between latent classes over time. If your point refers to something else or you would like us to state this in the manuscript, please let us know.

Comments 7: Abbreviations within tables should be explained as footnotes below the table.

Response 7:

Thank you for bringing this up. For Table 1, we now added the following text as a footnote on page 8, line 339:

“SD: Standard Deviation”

For table 2, we added the following footnote text on page 9, lines 361-362:

“*AIC: Akaike Information Criterion; BIC: Bayesian Information Criterion; aBIC: Adjusted Bayesian Information Criterion”

Comments 8: “Previous research has identified two types of smartphone use that are associated with habitual response patterns: process use and social use. While social-use was not found associated with problematic behaviors, like smartphone addiction, process-uses such as consuming media, browsing the internet, and playing games were associated with maladaptive outcomes [11].” When we say previous research, more than one study is highlighted but only one reference is shown. The same number of sources should be cited as the highlighted study.

Response 8:

Thank you for pointing out this inconsistency. We have now clarified this. Please see page 11, lines 421-422:

“One previous study has identified two types of smartphone use that are associated with habitual response patterns: process use and social use.”

Comments 9: Page 9, line 363-365, “Finally, we found that participants who did not identify as male or female reported higher peer attachment over time. Unfortunately, the sample size for this group was relatively small (N=87), which makes interpretation difficult.” Although a clear comment cannot be made about this result, the gender-related results of the studies in the literature should be given. Inferences can be made by avoiding definitive judgments in the comments.

Response 9:

Thank you for bringing this up. This was also mentioned by 2 other reviewers. To make the statistical findings more interpretable, we have now excluded adolescents who were in the “Other Sex” category. Now the sample (N=1303) only includes those who identify as either female and male. We have also updated all our results throughout the manuscript.

Comments 10: Page 10, line 385; In the conclusion section, recommendations for families, children, teachers, health professionals and all segments of society based on the clinical results of the study reveal the social contribution aspect of the study. Therefore, add beneficial recommendations related to these people to this section.

Response 10:

Thank you for this very valuable feedback. We now further added the following to the conclusion section to indicate the social contribution aspect of the study. Please see page 12, lines 476-478:

“More broadly, families, education centers, health professionals, and society could get involved to enable adolescents to critically evaluate their use of smartphones and other digital technologies.”

Round 2

Reviewer 3 Report

Comments and Suggestions for Authors

Dear author,

I understand that you have made changes to your article to improve it.

However, there is a key point that I still do not understand, and it seems to involve a confusion of concepts.

You are referring to research questions or objectives in your whole text as if they were the same concept; however, these two concepts are not the same and have different meanings. You must explicitly explain what you are referring to and revise your whole text.

In the conclusion, adding the phrase “This finding well-answered our three research questions” is insufficient. Once again, it shows the confusion of the two concepts (Research questions and objectives).

As I said before, the discussion is where you discuss your objectives and whether they were achieved with your literature revision. In the conclusion, you refer to your research question. And I will add again that your conclusion is very small and needs to be profoundly improved.

Author Response

Comment 1:

Dear authors,

I understand that you have made changes to your article to improve it. However, there is a key point that I still do not understand, and it seems to involve a confusion of concepts. You are referring to research questions or objectives in your whole text as if they were the same concept; however, these two concepts are not the same and have different meanings. You must explicitly explain what you are referring to and revise your whole text. In the conclusion, adding the phrase “This finding well-answered our three research questions” is insufficient. Once again, it shows the confusion of the two concepts (Research questions and objectives). As I said before, the discussion is where you discuss your objectives and whether they were achieved with your literature revision. In the conclusion, you refer to your research question. And I will add again that your conclusion is very small and needs to be profoundly improved.

Response 1:

Thank you for your thoughtful feedback. We carefully reviewed your comments and made revisions to the manuscript to ensure a clear distinction between our research question and research objectives throughout the text. We acknowledge that in the previous version, the two concepts were not consistently separated, which may have led to confusion. We have clarified the overarching research question in the introduction and discussion as the central guiding inquiry of the study, while ensuring that the research objectives are presented as the specific steps undertaken to investigate this question, please see page 4.

We also revised the discussion and have now structured our findings to explicitly address whether the research objectives were met, rather than implying that they directly answered the research question. The discussion now clearly outlines how each objective was examined and the extent to which the findings support them, ensuring that they remain distinct from the broader inquiry guiding the study. Additionally, we refined our language to prevent any confusion of these two concepts and conducted a thorough review of the entire manuscript to eliminate inconsistencies.

The conclusion has also been substantially revised to reflect these changes. We removed any references to research objectives in this section and instead focused on directly answering the research question based on our key findings. The revised conclusion now provides a more comprehensive synthesis of how habitual smartphone use influences adolescent peer attachment over time, situating our findings within the broader context of digital engagement and social relationships. We also expanded the discussion on the implications of our results, emphasizing both the risks and potential benefits of smartphone use, as well as the need for future research to explore the bidirectional nature of this relationship. By restructuring and expanding the conclusion, we have ensured that it fully addresses the research question rather than revisiting the research objectives, in line with your recommendation, please see page 13-14.